# Path Planning of Electric VTOL UAV Considering Minimum Energy Consumption in Urban Areas

**Yafei Li \*** 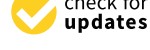 **and Minghuan Liu**

School of Air Traffic Management, Civil Aviation University of China, Tianjin 300300, China
\* Correspondence: commissioner@126.com; Tel.: +86-18622815319

**Abstract:** As a new mode of transportation in the future, electric vertical take-off and landing unmanned aerial vehicles (eVTOL UAV) can undertake the task of logistics distribution and carry people in urban areas. It is challenging to carry out research designed to plan the path of eVTOL UAVs which can have a safe and sustainable operation mode in urban areas. Therefore, this work proposes a method for planning an obstacle-free path for eVTOL UAVs in urban areas with the goal of minimizing energy consumption. It aims to improve the safety and sustainability of eVTOL UAV operations. Based on variations of air density with height, a more accurate formula for calculating battery energy consumption of eVTOL UAV is derived. It is used in the vertical takeoff and landing phase and horizontal flight phase, respectively. Considering the influence of buildings on eVTOL UAV operation, a path planning method applicable to complex urban environments is proposed. The safe nodes of eVTOL UAV flight are obtained by using Voronoi diagrams based on building locations. Then, the complete shortest and obstacle-free path is obtained by using a Dubins geometric path and Floyd algorithm. After obtaining the obstacle-free paths for all flight height zones, the battery energy consumption of the eVTOL UAV in each flight height zone is calculated. Then, the flight height with the minimum energy consumption is obtained. The simulation results show that the path length obtained by the proposed path planning method is shorter than that obtained by particle swarm optimization; the total battery energy consumption changes in the same pattern in the low-altitude areas and high-altitude areas; the difference between the maximum and minimum energy consumption in the small area enables the eVTOL UAV to cover about 350 m more, and about 420 m more in the large area. Therefore, in future high-frequency UAV mission flights, choosing the altitude with the lowest energy consumption can reduce UAV operator costs. It can also significantly increase UAV transport range and make UAVs operate more sustainably.

**Keywords:** air mobility; eVTOL UAV; path planning; minimum energy consumption; urban areas; flight height



## 1. Introduction

In recent years, urbanization has been advancing, urban populations are growing rapidly, and traffic congestion caused by high population concentrations has become a major problem for urban governance. With the promotion of 5G information technology and breakthroughs in technologies such as autonomous driving and electric propulsion [1], electric vertical take-off and landing unmanned aerial vehicles (eVTOL UAV) have been rapidly developed, which has brought urban air mobility (UAM) back into the limelight. UAV is a safe and sustainable advanced air transport system for the transportation of passengers and cargo in urban areas [2]. Using eVTOL UAV as transportation vehicles, urban air transport is three-dimensional, efficient, environmentally friendly, has a low-cost, and can complement and connect with ground transportation to effectively reduce ground traffic congestion [3]. As the main means of transport for urban air traffic, eVTOL UAVs are also an emerging means of transport with broad application prospects.

EVTOL UAVs are aerial vehicles with electric power system. Unlike small rotary-wing UAVs, most eVTOL UAVs are mainly rotary-wing and fixed-wing composites. In the take-off and landing phases, eVTOL UAVs rely on rotor blades to complete vertical take-off and landing, and in the horizontal flight phase, they rely on fixed-wing to complete flight [4], which has the characteristics of fast speed, a long range, and large load of fixed-wing UAVs. Electric vertical take-off and landing technology is still in the research stage, and eVTOL UAVs have therefore received widespread attention from the aviation community. EVTOL UAVs have the ability to hover in the air [5] and have good maneuverability since they can take off and land vertically. They also occupy a small landing area and do not rely on runways for take-off and landing. As an electrically propelled aircraft, eVTOL UAVs have zero emission potential and are well suited for UAM. The use of eVTOL UAVs by logistics companies to carry out logistics and distribution tasks in cities can reduce transport costs and delivery times, as well as effectively alleviate traffic congestion on the ground.

The rapid development of UAV technology has made it possible for a large number of UAVs to enter urban low-altitude airspace, and the high-density UAV traffic flow in the future poses a challenge to the existing air traffic management system [6,7]. In order to make UAVs have an efficient, orderly, sustainable, and safe operation mechanism, the research of UAS air traffic management (UTM) problem has received extensive attention from scholars at home and abroad. The traditional air traffic management system is not suitable for UAS air traffic management due to various factors such as the complexity of urban low-altitude airspace, the variety of UAS, different mission loads, and the fact that the existing communication and navigation surveillance systems cannot be directly applied to UAS [7,8]. Therefore, some countries with more developed aviation industries are studying how to establish a set of air traffic management rules that can be applied to UAVs, and actively promote the establishment of an air traffic management system for UAVs. At present, countries and regions such as the United States, Europe, China, Singapore, and Japan have all initially established their own UTM management frameworks [9]. In these management frameworks, the issue of traffic safety remains the primary issue of UAV traffic management, and issues related to the safe operation of UAVs in urban areas have become a hot spot for research. Avoiding ground obstacles that affect the safety of UAV operations and other aircraft that may conflict with them are key aspects of ensuring safe UAV operations. Assessing the risks of UAV operations and planning UAV-free paths are reliable ways to ensure safe UAV operations, improve the viability of UAV urban low-altitude airspace and enhance UAV risk management.

In the study of safety issues for UAV operations in urban areas, numerous researchers have planned UAV paths to improve UAV survivability. Ramana et al. [10] considered obstacles as hindrances to UAV operations and studied the impact of urban obstacles on UAV flight paths; Filippis et al. [11] applied the Theta* algorithm in 3D path planning to terrain obstacles and urban environments to evaluate solutions for different types of obstacles; He et al. [12] designed a new multi-stage dynamic path planning algorithm that combines an artificial potential field method with a reconciliation function, Kalman filter, and Markov decision process, which is mainly applicable in dynamic 3D environments shared by multiple aircraft with potential conflict risks for applications in urban air traffic; Wu et al. [13] proposed a risk-bounded path planning algorithm for UAVs in dynamic and uncertain urban environments that can generate collision-free paths with guaranteed probability; Wu et al. [14] investigated an improved fast randomly expanding spanning tree algorithm for online obstacle avoidance trajectory planning of multi-rotor UAVs for discrete urban environments, enabling UAVs to adapt to dynamic changes in urban environments; Munoz et al. [15] developed two overlay path algorithms for urban environments that have obstacle avoidance capabilities and provide safe and smooth paths; Honghong Zhang et al. [16] performed model simplification and risk assessment for complex low-altitude urban environments to obtain a low-altitude 3D risk map, then used the risk value as the path cost and applied an improved ant colony algorithm to obtain a less costly path;

Majeed et al. [17] in the presence of multiple obstacles in 3D urban environments, planning minimum length, collision-free and flyable paths to cover spatially distributed areas.

The aforementioned literature has studied the risk of UAV operation in urban areas, analyzed the impact of urban obstacles on UAV operation, constructed a model to assess the risk value of UAV paths, and used algorithms to plan optimal UAV paths. However, less consideration has been given to the issue of battery consumption for UAV operation in realistic situations. Notably, aiming for the lowest energy consumption is one of the challenges of path planning [18] for the following reasons: lithium batteries are the main energy supply unit for eVTOL UAVs and can only supply a limited amount of power and energy [19], and excessive battery consumption becomes a major cause of UAV crashes [20]. Battery energy consumption is affected by flight time, speed, load, and other factors [21,22]. In the UAV logistics distribution problem, the factors such as speed, load and path length are generally considered, and the energy consumption model is constructed, then a solution strategy is proposed to reduce the energy consumption and operation cost [23–25]. EVTOL UAVs are mainly used for short-range urban flights, and prolonged flights make excessive battery depletion a significant risk challenge [26]. At the same time, UAV battery energy consumption also limits the coverage of UAVs [27] and affects the cost of using them. In order to ensure the safe operation of UAVs in urban areas while taking into account the low energy consumption of urban air traffic, it is of great practical importance to analyze and study the energy consumption law of UAV operation in order to achieve minimal operational energy consumption. UAV energy consumption is related to the length of the UAV flight path, and path planning is the process of planning the optimal path from the starting point to the target point, taking into account various factors in the operating environment. Planning an obstacle-free path and finding the path with the minimum energy consumption can be considered at the same time [28].

Cities represent an area of concentrated buildings, and these buildings have heights that occupy a lot of space. In mega cities, some buildings are over 100 m in height. However, 96.5% of UAVs currently operate in low-altitude airspace below 120 m in height [29]. Therefore, the primary threat to the safety of UAV operations in urban areas comes from urban buildings, and the primary consideration for path planning is the impact of buildings on the safe operation of UAVs [30]. As buildings vary in height, the number of buildings affected by UAVs at different flight heights varies. The UAV can fly over buildings that are lower than its height, and the higher the flight height, the fewer high-rise buildings it needs to avoid, but higher flight heights will consume more batteries. In order to minimize energy consumption and increase endurance, it is not necessary to place the minimum cruise altitude above the height of all urban buildings for electric vertical take-off and landing aircraft, and a lower altitude can be chosen to fly laterally around a small number of over-height buildings [31]. Therefore, the impact of buildings on the UAV at different flight heights in urban low-altitude airspace must be considered when planning the path with the goal of minimizing energy consumption.

The main existing UAV path planning techniques include heuristic algorithms, potential field methods, and geometric methods [18]. Heuristic algorithms include ant colony algorithms, genetic algorithms, particle swarm optimization, and others. These algorithms have high complexity and do not necessarily obtain the optimal solution in the process of finding the best solution. The potential field methods often fall into local optimality and fail to reach the destination. The advantage of the geometric methods is that the path obtained is more accurate. This study proposes an accessible path planning method for urban areas, combining the improved Voronoi diagram method with Dubins geometric curves, and finally using the Floyd algorithm to obtain the shortest and most obstacle-free path.

In view of this, this paper takes eVTOL UAVs as the research object, aims at minimizing energy consumption and considers the influence of urban buildings on UAV operations at different flight heights when planning paths. With a view to planning an obstacle-free path for UAV operations in urban areas and with minimal energy consumption, this paper

provides a feasible UAV control decision to promote the healthy development of UAV air traffic management.

The rest of the manuscript is organized as follows. Section 2 presents a method for calculating the battery output power and energy consumption, taking into account the effect of air density. Section 3 presents a path planning method using Voronoi diagrams, Dubins geometry curves, and the Floyd algorithm in combination with each other. Section 4 presents the mechanism of the influence of urban buildings on UAV flight height. In Section 5, the change in energy consumption of the UAV is analyzed through arithmetic simulation to obtain the path with the minimum energy consumption. The path planning method proposed in this paper is compared with the particle swarm optimization to highlight the advantages of the path planning method in this paper. Finally, Section 6 presents the conclusion of this paper and provides an outlook for future work.

## 2. Battery Output Power and Calculation of Energy Consumption

### 2.1. Battery Output Power

From a flight profile perspective, the entire flight process of an electric vertical takeoff and landing UAV performing a mission in an urban area can be divided into a horizontal flight phase and a vertical flight phase.

In the horizontal flight phase, the eVTOL UAV mainly relies on the fixed wing to achieve smooth flight, and the aerodynamic layout is similar to that of a fixed-wing aircraft. The eVTOL UAV electric thrust mainly overcomes the air resistance during flight, and its battery output power $P_1$, that is [32]:

$$P_1 = \frac{1}{\eta_P \eta_M \eta_{ESC}} \left( \frac{1}{2} \rho V_C{}^3 S C_{D0} + \frac{2kM^2 g^2}{\rho S V_C} \right) \tag{1}$$

where $S$ is the horizontal windward area of the UAV; $C_{D0}$ is the zero-lift drag coefficient; $\rho$ is the air density; $V_C$ is the horizontal cruising speed; $k$ is the induced drag factor; $M$ is the total takeoff mass of the UAV; $g$ is the acceleration of gravity, taken as 9.8 m/s$^2$; $\eta_P$, $\eta_M$, and $\eta_{ESC}$ are the efficiency of the propeller, brushless motor and brushless electronic speed control, respectively, which are generally taken as 0.8.

In the vertical flight phase, the electric vertical take-off and landing UAV relies on rotor flight. During the slow uniform flight, the electric thrust mainly overcomes the weight of the UAV and the battery output power $P_2$ can be expressed as [32]:

$$P_2 = \frac{1}{\eta_P \eta_M \eta_{ESC}} \sqrt{\frac{(Mg)^3}{2\rho A \kappa}} \tag{2}$$

where $A$ is the area of the UAV paddle disk, the reference horizontal windward area in this paper; $\kappa$ is the correction factor, generally taken as 0.9~0.94.

### 2.2. Energy Consumption Calculation

Considering that the air density in the standard state is related to the height from the sea level, the battery output power of the electric vertical takeoff and landing UAV in the horizontal flight phase and vertical flight phase is related to the air density according to Equations (1) and (2). The battery output power of the electric vertical takeoff and landing UAV varies to a certain extent at different flight heights. Considering the effect of flight height on the battery output power is more suitable for the actual operating condition of the UAV and can solve the battery energy consumption more accurately.

According to the BADA manual, the air density at standard atmospheric sea level $\rho_0$ is 1.225 kg/m$^3$ and the temperature at standard atmospheric sea level $T_0$ is 288.15 K.

At an altitude of $H$ ($H < 11{,}000$ m) from sea level, the temperature $T$ and air density $\rho$ decrease with increasing altitude with the following pattern [33]:

$$T = T_0 - 0.0065\,H \tag{3}$$

$$\rho = \rho_0 \left( \frac{T}{T_0} \right)^{\frac{-g}{K_T R} - 1} \tag{4}$$

where $R$ is the gas constant of air with a value of 287.05287 m$^2$/Ks$^2$; $K_T$ is the temperature gradient below the top of the troposphere with a value of $-0.0065$ °K/m; $g$ is the acceleration of gravity and is taken as 9.8 m/s$^2$.

Figure 1 reflects the situation of the eVTOL UAV flying at a ground altitude $h_1$ and the flight height $h_2$.

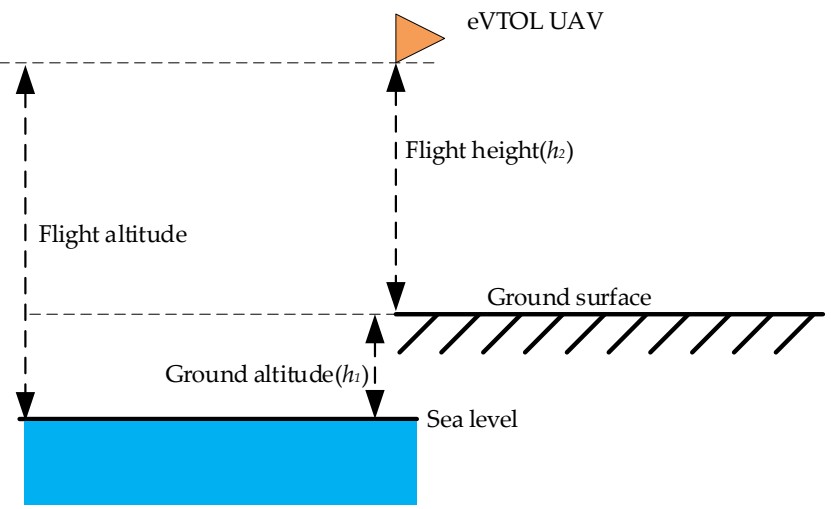

**Figure 1.** The situation of the eVTOL UAV.

Combining Equations (1), (3) and (4), the energy consumed by the battery $Q_1$ during the horizontal flight phase of the eVTOL UAV at a horizontal flight distance $L$ can be expressed as:

$$\begin{cases} Q_1 = \frac{L}{V_C \eta_P \eta_M \eta_{ESC}} \left\{ \frac{1}{2} \rho V_C^3 S C_{D0} + \frac{2kM^2 g^2}{\rho S V_C} \right\} \\ \rho = \rho_0 \left[ \frac{T_0 - 0.0065(h_1 + h_2)}{T_0} \right]^{\frac{-g}{K_T R} - 1} \end{cases} \tag{5}$$

Combined with Equations (2)–(4), the ground altitude in the flight area is $h_1$ and the flight height is $h_2$. When the speed of the eVTOL UAV in the vertical takeoff speed is $V_T$ and the speed in the vertical landing flight speed is $V_L$, the battery energy consumption $Q_2$ for vertical takeoff and $Q_3$ for vertical landing can be expressed as:

$$Q_2 = \int_0^{\frac{h_2}{V_T}} \left\{ \frac{1}{\eta_P \eta_M \eta_{ESC}} \sqrt{\frac{(Mg)^3}{2\rho_0 \left[ \frac{T_0 - 0.0065(h_1 + V_T t)}{T_0} \right]^{\frac{-g}{K_T R} - 1} A\kappa}} \right\} dt \tag{6}$$

$$Q_3 = \int_0^{\frac{h_2}{V_L}} \left\{ \frac{1}{\eta_P \eta_M \eta_{ESC}} \sqrt{\frac{(Mg)^3}{2\rho_0 \left( \frac{T_0 - 0.0065(h_1 + h_2 - V_L t)}{T_0} \right)^{\frac{-g}{K_T R} - 1} A\kappa}} \right\} dt \tag{7}$$

Then the energy consumed by the battery $Q$ during the whole flight process of the eVTOL UAV can be expressed as:

$$Q = Q_1 + Q_2 + Q_3 \tag{8}$$

From the above equation, it can be seen that the battery energy consumption of electric vertical takeoff and landing UAV in the vertical flight phase increases with its flight height and also with the local altitude, and the battery energy consumption in the horizontal

flight phase increases with the increase of the horizontal flight path length. In urban environments, the number and location of buildings to be avoided by UAVs at different flight heights are different due to the varying heights of buildings, and the horizontal flight path length decreases with increasing altitude. The battery energy consumption of UAVs in both horizontal and vertical flight phases varies at different flight heights. Under these conditions, exploring the effects of UAV flight height, local altitude and horizontal flight path length on battery energy consumption, and optimizing UAV flight height and horizontal flight path with the goal of minimizing energy consumption, can provide a decision basis for UAV to choose the operation mode with minimum energy consumption.

## 3. Safe Horizontal Path Planning Method

The line segment in the traditional Voronoi diagram method is a mid-pipeline between threat points, and the generated paths are safe since they are far from obstacles [34]. UAV path planning using the Voronoi diagram method usually considers the threat as a point in the two-dimensional plane and the mid-pipeline as the UAV flight path can make full use of airspace resources and establish a safe path with minimal threat cost [35,36]. The location, number and height of urban buildings are constant (i.e., the Voronoi diagram method is suitable for planning horizontal flight paths of UAVs in urban areas when the threat source distribution is known). However, since buildings have a certain width, treating them as a threat point in space and ignoring the space occupied by buildings can lead to potential huge safety hazards in the UAV path. The solution could be to consider the building as a circular boundary and apply Dubins geometric curves to plan a smooth shortest path [37]. The Voronoi diagram method is combined with Dubins geometric path to plan a safe UAV path with the characteristics of Dubins path smoothing while considering the influence range of the building. Finally, Floyd algorithm is applied to solve the global shortest path. This new UAV path planning model has the advantages of both safety and having shortest distance, and is suitable for UAV operation in urban environments.

### 3.1. Generation of Safe Nodes

Considering that the building has a certain shape, the building cannot be directly equated to a point. In order to better evaluate the threat range of buildings, buildings are abstracted into an external circle of buildings, the center of the circle is the center point of the buildings, and the radius of the circle is the farthest distance from the center point of the buildings to the boundary [38]. When multiple buildings are similar in height and close in distance, multiple buildings can be considered as the same external circle; the model is shown in Figure 2.

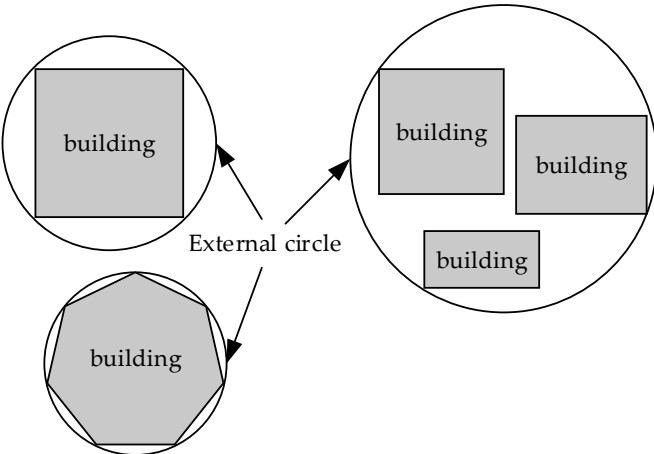

**Figure 2.** The external circle of buildings.

The coordinates of the circle center of the external circle can be obtained when the location of the buildings is known. The center of the circle is considered as the source point of threat, and the safe path that the UAV can fly can be obtained by Voronoi diagram if the distribution of threat source point is known, as shown in Figure 3.

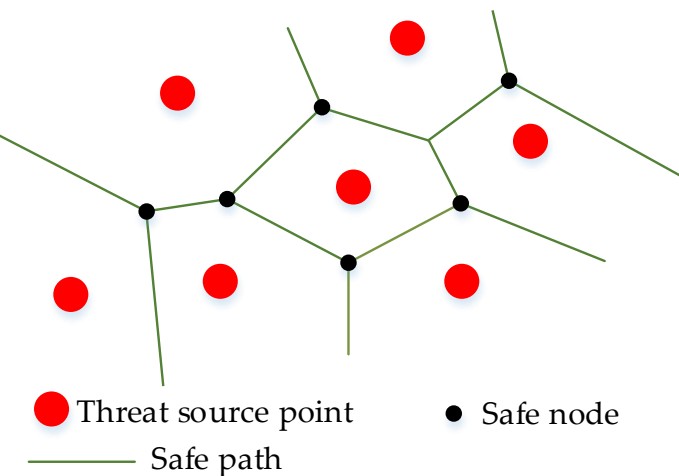

**Figure 3.** Voronoi diagram.

The line segment is the flyable safe path. The intersection of line segments and line segments (i.e., the small black dots in the figure) constitute the safety nodes of the UAV path, and there are several safety nodes around each threat source point, and only the adjacent safety nodes are connected to each other by line segments. The coordinates of each safe node can be found with the known distribution of threat source. The Voronoi diagram method can be used to obtain all of the reachable safety nodes of the UAV while making full use of the airspace resources.

### 3.2. Dubins Geometric Path

The traditional Voronoi diagram method let the UAV fly along the line segment in the diagram, passing through a number of adjacent safe nodes to ensure the least cost of being affected by the threat, but can't meet the objective of shortest path length. In the actual UAV flight, all safety nodes can pass each other, and considering that the UAV can fly between any two non-adjacent safety nodes can avoid the path being too long due to too many twists and turns between the starting and ending points, as shown in Figure 4.

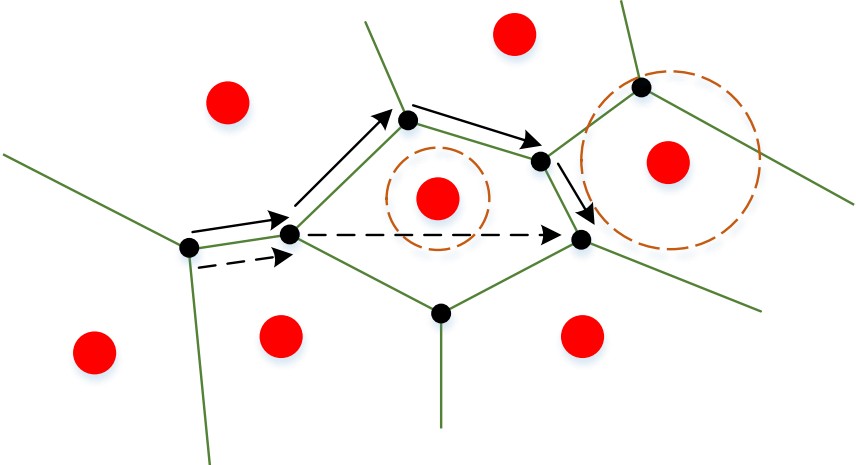

**Figure 4.** Improved path planning.

The solid line with arrows in the figure is the conventional path, and the dashed line with arrows is the path containing non-adjacent safety nodes; the flight path along the dashed line is shorter. In order to achieve the goal of minimum energy consumption, the path should be planned in such a way that the path is as short as possible. Considering the influence of buildings on UAV operation, buildings cannot be simply regarded as a point, and the risk of UAV operation is extremely high in the threat circle formed by the outer circle of buildings.

The influence of buildings on the path is reflected in two aspects.

1. In the Voronoi diagram, the lines between some non-adjacent safe nodes may be in the threat circle domain. The left dashed circle in Figure 4 is the external circle of the building, and the dashed line passes through the outer circle, indicating that it is extremely risky for the UAV to fly in a straight line between these two points.

2. Some safe nodes may be influenced by buildings. The right dashed circle in Figure 4 contains a safety node, indicating that the safety node is located in the risky circle domain and should be discarded when planning the path.

For the first influence, using Dubins geometric path idea, the UAV should avoid the risk circle domain when flying, the UAV obstacle avoidance path consists of the tangent line of the circle and the arc, the path length is the shortest under the condition of ensuring safety, and the UAV bypasses the obstacle as shown in Figure 5.

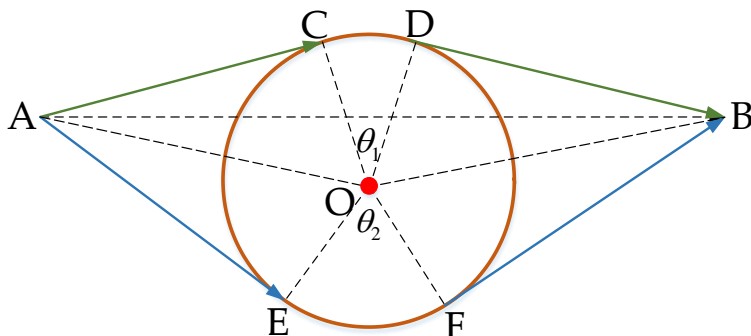

**Figure 5.** Dubins geometry path.

The circle in the figure is the external circle of the building, the center of the circle is the point O, the radius of the circle is $r$, the angle between OC and OD is $\theta_1$, the angle between EO and FO is $\theta_2$, point A is the starting point of the UAV flight, point B is the target point of the UAV. Line AB passes through the danger zone, the flight path of the drone must avoid the circle under the condition of ensuring safety, but also to make the path shortest.

The line AC, DB, AE, FB are tangents to the circle, and the center of the circle O is below AB. Now there are two paths: AC→arc CD→DB and AE→arc EF→FB. Since O is below AB, $\theta_2$ is greater than $\theta_1$, AC→arc CD→DB is the shorter path.

It can be concluded that, in general, the shortest path consisting of a tangent line and an arc should be on the same side of the center of the circle as the base line, using the line connecting the starting point and the target point as the base line. Assuming that the length of AB is, the length of AO is, the length of OB is, the modified Dubins geometric path length X can be expressed as:

$$X = \sqrt{d_1^2 - r^2} + \sqrt{d_2^2 - r^2} + r[\arccos(\frac{l^2 - d_1^2 - d_2^2}{2}) - \arccos(\frac{r}{d_1}) - \arccos(\frac{r}{d_2})] \quad (9)$$

If there are multiple buildings between safe nodes, the UAV needs to fly around more times, has poor maneuverability and greater flight risk, and it is more difficult for ground personnel to operate. In order to improve the survivability of the UAV and reduce the risk of UAV operation, the UAV is not considered to fly between multiple buildings for the time being, and the UAV can only fly when there is at most one building between the safe nodes.

### 3.3. The Shortest Path Planning Method Based on Floyd Algorithm

The Floyd algorithm is an algorithm used to find the shortest path between multiple source points in a given weighted graph. In the case of known start and end point locations, the safety node can be obtained after applying Voronoi diagram, and the shortest distance between any two points including start point, end point and safety node after applying Dubins geometry path. The connection between the starting point and the end point may pass through several buildings, and the UAV cannot fly directly and needs to pass through several safety nodes to complete the flight mission. The nodes and the connecting lines between points form a network diagram, which is shown in Figure 6.

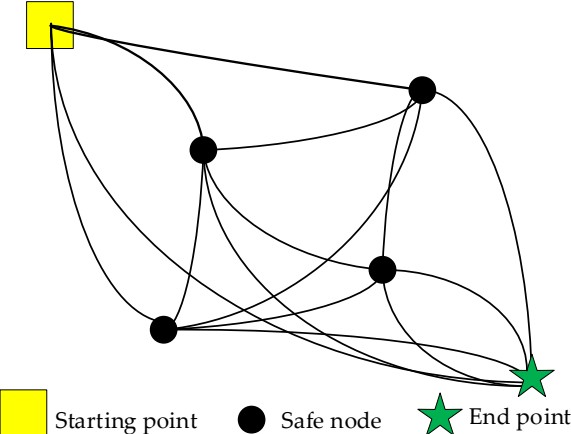

**Figure 6.** Network diagram.

The starting point, the end point and the safety nodes in the figure constitute the points in the network diagram, and the distance between two points is the weight between the two points. If there are multiple buildings between two points, the distance between the two points is set to infinity. According to the generated UAV path nodes, the shortest distance from the starting point to the end point can be solved. The Floyd algorithm can solve the shortest distance between any two nodes in the network graph, and can find the shortest path between the starting point→safety node→end point, and finally obtain a safe and feasible horizontal flight path of the UAV with the shortest distance.

## 4. Optimum Flight Height

### 4.1. Division of the Flight Height Zones

In the flight area, there are a number of buildings from the starting point to the end of the UAV flight. Dividing the UAV flight height interval based on the building height data is a prerequisite to ensure the safe operation of the UAV. A rectangular area including a straight line connecting the starting point to the end point of the UAV is defined as the UAV flight area. Within the flight area, all building heights are obtained. All building heights are arranged from small to large, and every two adjacent heights form a flight height interval for the UAV, which eventually forms several flight height intervals. The division of flight height intervals is shown in Figure 7.

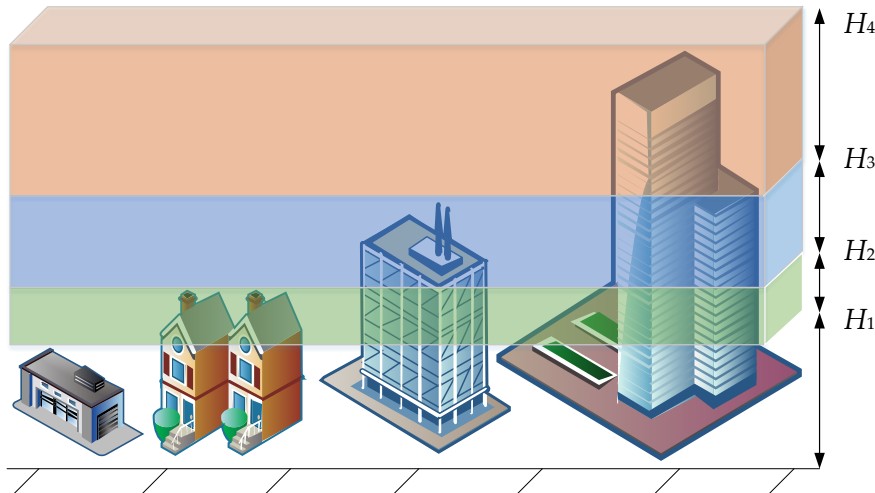

**Figure 7.** Division of flight height zones.

There are buildings of different heights in the figure, and from small to large the heights are $H_1$, $H_2$, $H_3$, and $H_4$. According to the height of the building divided into different flight height zones, from the low to high perspective, the three flight height zones height range in the figure are $(H_1, H_2)$, $(H_2, H_3)$, $(H_3, H_4)$.

### 4.2. The Flight Height with the Minimum Energy Consumption

The flight height range according to the building height is characterized by:

1. When the UAV is in the flight height zone $(A, B)$, it can fly over buildings with height no higher than $A$, and it needs to bypass buildings with height no lower than $B$;
2. In the same flight height range, when the flight height of the UAV changes within the range, the number and location of buildings affecting the flight safety of the UAV remain unchanged, and the safe horizontal path remains unchanged;
3. In different flight height zones, the number and location of buildings affecting the safety of UAV flight are different, and the safe horizontal path is also different;
4. In the lower flight height range, there are also more buildings that affect the safety of UAV flight.

According to Equations (6) and (7), it can be seen that the energy consumption in the vertical takeoff and landing phase increases with the increase of altitude. In the same flight height interval, the horizontal flight path remains the same and the energy consumption in the horizontal flight phase remains the same. To minimize energy consumption, the UAV flight height is set to the minimum value of the flight zone height. For example, when the flight height zone is $(A, B)$, the UAV flight height is set to $A$, and the total energy consumption of a flight height zone can be solved when the height is known. All of the UAV flight heights mentioned in this paper refer to the height of the UAV from the ground. According to Equations (5)–(7), the air density is affected by altitude, and the local altitude should be considered for calculating the energy consumption. The total energy consumption of the lowest altitude in all flight height zones is calculated, and finally the optimal flight height with the minimum total energy consumption is obtained. The logical framework of this study is shown in Figure 8.

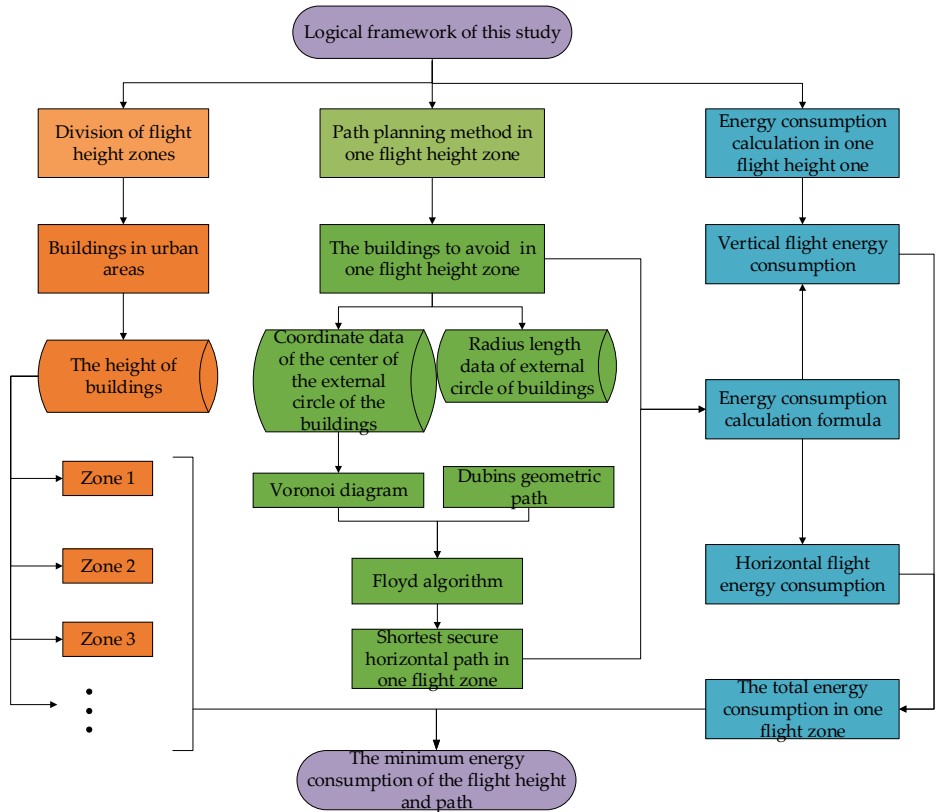

**Figure 8.** Logical framework of this study.

## 5. Simulation Analysis and Discussion

### 5.1. Flight Area Settings

With reference to relevant data in reference [39], the parameters of eVTOL UAV are shown in Table 1.

**Table 1.** Parameters of eVTOL UAV.

| Parameter | Value | Parameter | Value |
|---|---|---|---|
| $M$ | 6.2 Kg | $\kappa$ | 0.94 |
| $A$ | 1.313 m$^2$ | $\eta_P$ | 0.8 |
| $V_C$ | 15 m/s | $\eta_M$ | 0.8 |
| $V_T$ | 5 m/s | $\eta_{ESC}$ | 0.8 |
| $V_L$ | 5 m/s | $C_{D0}$ | 0.015 |
| $k$ | 0.13 | $g$ | 9.8 m/s$^2$ |

An urban area is selected, and the regional model is shown in the figure. The flying area of the eVTOL UAV is rectangular with a length of 2200 m and a width of 730 m. The starting point coordinate (0, 730) and the end point coordinate (2200, 0) are set. The building parameters and numbers are set according to the location and size of the buildings in the Figure 9. The building parameters are shown in Table 2.

The circles in Figure 8 represent external circles of buildings. The numbers in Figure 8 are serial number of buildings.

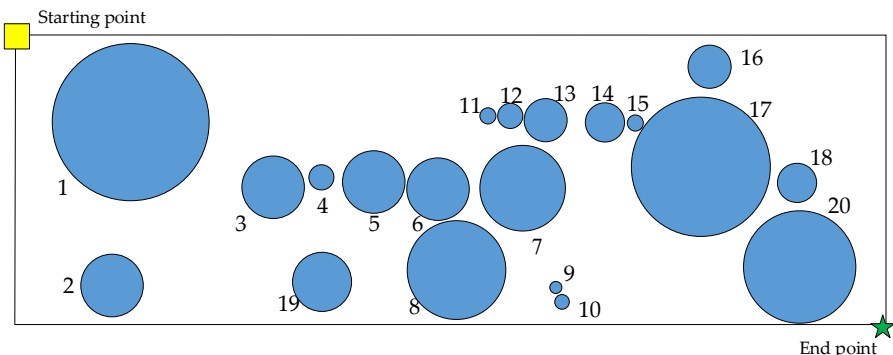

**Figure 9.** Flight area.

**Table 2.** Parameters of external circles of buildings.

| Serial Number | Center Coordinates (m) | Circle Diameter (m) | Height of Building (m) | Serial Number | Center Coordinates (m) | Circle Diameter (m) | Height of Building (m) |
|---|---|---|---|---|---|---|---|
| 1 | (286, 528) | 403 | 30 | 11 | (1185, 543) | 40 | 10 |
| 2 | (237, 114) | 160 | 40 | 12 | (1245, 541) | 60 | 20 |
| 3 | (644, 365) | 160 | 40 | 13 | (1335, 531) | 110 | 30 |
| 4 | (767, 387) | 70 | 10 | 14 | (1485, 525) | 100 | 30 |
| 5 | (899, 376) | 160 | 50 | 15 | (1560, 523) | 40 | 10 |
| 6 | (1060, 355) | 160 | 30 | 16 | (1747, 666) | 110 | 10 |
| 7 | (1276, 357) | 220 | 20 | 17 | (1726, 414) | 350 | 50 |
| 8 | (1108, 151) | 250 | 50 | 18 | (1968, 373) | 100 | 10 |
| 9 | (1360, 109) | 30 | 10 | 19 | (770, 124) | 150 | 30 |
| 10 | (1375, 73) | 40 | 20 | 20 | (1963, 144) | 286 | 60 |

### 5.2. Horizontal Path Planning

5.2.1. Planning Path by Method in This Manuscript and PSO

In order to show that the path planning method proposed in this study has the characteristics of shorter distance and global optimum, the path planning method in this study is compared with the particle swarm optimization (PSO), and the path length obtained by the two methods is compared.

Particle swarm optimization (PSO) can solve the UAV path planning problem. The core idea of PSO is the update of particle velocity and position, and the update formula is as follows:

$$X^{k+1} = X^k + V^{k+1} \tag{10}$$

$$V^{k+1} = \omega V^k + c_1 r_1 (P_{id}^k - X^k) + c_2 r_2 (P_{gd}^k - X^k) \tag{11}$$

where $k$ is the number of iterations;

$X^{k+1}$ is the position of particle $i$ of the $k + 1$ generation;

$X^k$ is the position of particle $i$ of the $k$ generation;

$V^{k+1}$ is the velocity of particle $i$ of the $k + 1$ generation;

$V^k$ is the velocity of particle $i$ of the k generation;

$P_{id}^k$ is the position of the individual optimal particle $i$ of the $k$ generation;

$P_{gd}^k$ is the position of the global optimal particle of the $k$ generation;

$r_1$ and $r_2$ are random numbers between (0, 1);

$c_1$ and i $c_2$ are constants.

The velocity and position of the particle are not infinite, the velocity and position of the particle are limited. As far as path planning is concerned, the limit range of particle

position is $x \in [-10, 10]$, and the limit range of particle velocity is $y \in [-10, 10]$, and the formula for calculating the speed limit range is as follows:

$$\begin{cases} v_x^{\max} = 0.1 * (x_{\max} - x_{\min}) = 0.1 * 20 = 2 \\ v_x^{\min} = -v_x^{\max} = -2 \\ v_y^{\max} = 0.1 * (y_{\max} - y_{\min}) = 0.1 * 20 = 2 \\ v_y^{\min} = -v_y^{\max} = -2 \end{cases} \quad (12)$$

The steps for PSO path planning are as follows:

STEP 1: create a two-dimensional environment.

STEP 2: Initialize parameters.

STEP 3: Initialize the position and velocity of each particle in the particle swarm.

STEP 4: In the main loop, iterate MaxIt times, and update the velocity and position of nPop particles each time. Take updating the velocity and position of a particle as an example.

STEP 4.1: Update the current particles $V_x^{k+1} = \omega V_x^k + c_1 r_1 (P_{idx}^k - X^k) + c_2 r_2 (P_{gdx}^k - X^k)$, and process the $V_x^{k+1}$ that are not in the speed range to make them in the range of $[-2, 2]$.

STEP 4.2: Update the current particles $X^{k+1} = X^k + V_x^{k+1}$, and process the $X^{k+1}$ that are not in the speed range to make them in the range of $[-10, 10]$.

STEP 4.3: Update the current particles $V_y^{k+1} = \omega V_y^k + c_1 r_1 (P_{idy}^k - X^k) + c_2 r_2 (P_{gdy}^k - X^k)$, and process the $V_y^{k+1}$ that are not in the speed range to make them in the range of $[-2, 2]$.

STEP 4.4: Update the current particles $Y^{k+1} = Y^k + V_y^{k+1}$, and process the $Y^{k+1}$ that are not in the speed range to make them in the range of $[-10, 10]$.

STEP 4.5: evaluate the objective function value of the particle.

STEP 4.6: update the individual optimal particle and global optimal particle of the current particle.

STEP 5: Output the global optimal particle.

According to the building height, the flight height zones are divided into (10 m, 20 m), (20 m, 30 m), (30 m, 40 m), (50 m, 60 m), corresponding to the flight height above the ground is 10 m, 20 m, 30 m, 40 m, 50 m, respectively. Find the paths at each of the five heights.

We use PSO to plan the eVTOL UAV path in the same scenario. There are many obstacle-free paths for drones from the start to the end, and our goal is to find the one with the shortest distance from the many routes. Using PSO, an obstacle-free path with the shortest distance can be found.

1.  The same method is used to divide the flight zones of the eVTOL UAV. According to the building height, the flight height zone is divided into (10 m, 20 m), (20 m, 30 m), (30 m, 40 m), (50 m, 60 m), corresponding to the flight height above the ground is 10 m, 20 m, 30 m, 40 m, 50 m, respectively. Find the paths at each of the five heights. Flight zone (10 m, 20 m);

In the flight zone of (10 m, 20 m), the flight height is 10 m, the buildings no higher than 10 m will be abandoned, and these buildings are numbered 4, 9, 11, 15, 16, 18.

The Voronoi diagram is used to generate 14 safe nodes. These points and their corresponding coordinates are shown in Table 3.

**Table 3.** Safe nodes at the flight height of 10 m.

| Safe Node | Coordinate (m) | Safe Node | Coordinate (m) |
|:---:|:---:|:---:|:---:|
| $a_1$ | (400, 305) | $h_1$ | (973, 167) |
| $b_1$ | (503, 138) | $i_1$ | (1405, 410) |
| $c_1$ | (776, 281) | $j_1$ | (1510, 279) |
| $d_1$ | (954, 190) | $k_1$ | (1318, 212) |
| $e_1$ | (1011, 589) | $l_1$ | (1281, 452) |
| $f_1$ | (1167, 433) | $m_1$ | (1517, 278) |
| $g_1$ | (1169, 282) | $n_1$ | (1667, 123) |

The horizontal path obtained by Dubins geometry path and Floyd algorithm is shown in Figure 10.

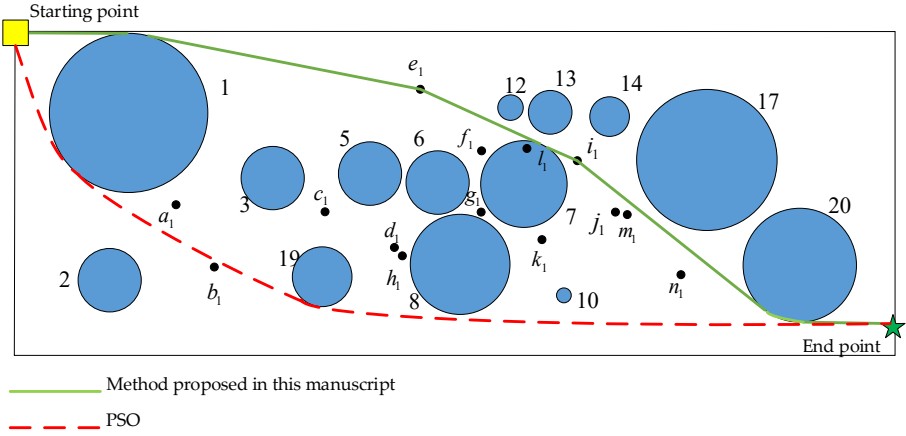

**Figure 10.** The horizontal path at the flight height of 10 m.

Finally, we can obtain the shortest horizontal path: starting point $\to e_1 \to i_1 \to$ end point. The shortest horizontal path length is 2392.82 m.

2. Flight zone (20 m, 30 m);

In the flight zone of (20 m, 30 m), the flight height is 20 m, the buildings no higher than 20 m will be aban-doned, and these buildings are numbered 4, 7, 9, 10, 11, 12, 15, 16, 18.

The Voronoi diagram is used to generate 10 safe nodes. These points and their corresponding coordinates are shown in Table 4.

**Table 4.** Safe nodes at the flight height of 20 m.

| Safe Node | Coordinate (m) | Safe Node | Coordinate (m) |
|---|---|---|---|
| $a_2$ | (400, 305) | $f_2$ | (1265, 37) |
| $b_2$ | (503, 138) | $g_2$ | (973, 167) |
| $c_2$ | (776, 281) | $h_2$ | (1571, 39) |
| $d_2$ | (954, 190) | $i_2$ | (1400, 282) |
| $e_2$ | (1028, 708) | $j_2$ | (1487, 212) |

The horizontal path obtained by Dubins geometry path and the Floyd algorithm is shown in Figure 11.

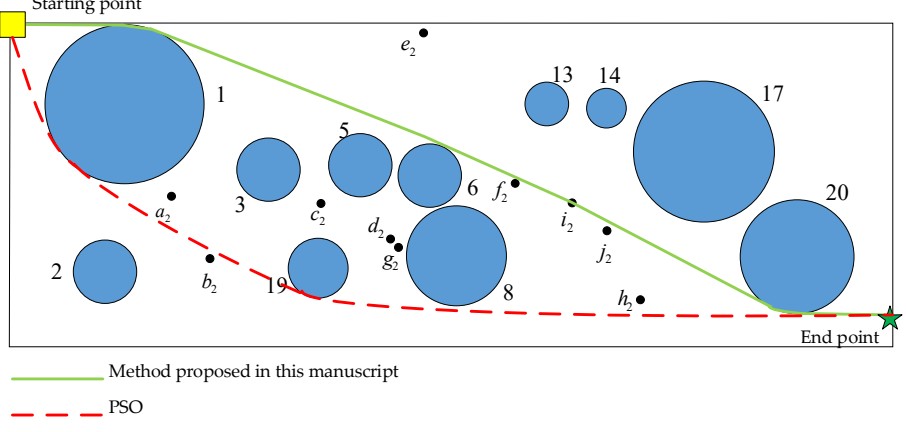

**Figure 11.** The horizontal path at the flight height of 20 m.

Finally get the shortest horizontal path: starting point $\rightarrow i_2 \rightarrow$ end point. The shortest horizontal path length is 2553.4 m.

3.  Flight zone (30 m, 40 m);

In the flight zone of (30 m, 40 m), the flight height is 30 m, the buildings no higher than 30 m will be abandoned, and these buildings are numbered 1, 4, 6, 7, 9, 10, 11, 12, 13, 14, 15, 16, 18, and 19.

The Voronoi diagram is used to generate two safe nodes. These points and their corresponding coordinates are shown in Table 5.

**Table 5.** Safe nodes at the flight height of 30 m.

| Safe Node | Coordinate (m) | Safe Node | Coordinate (m) |
| --- | --- | --- | --- |
| $a_3$ | (1304, 592) | $b_3$ | (1571, 39) |

The safe node $c_3$ formed by buildings numbered 2, 3, 5 are located outside the rectangular area. Since the UAV only flies in the rectangular area, the path length will inevitably increase when it flies outside the area, so safe node $c_3$ will be discarded. The horizontal path is shown in Figure 12.

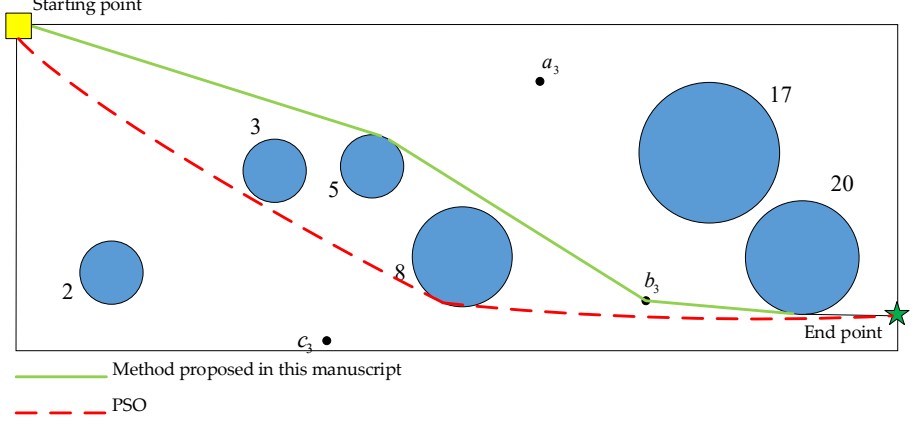

**Figure 12.** The horizontal path at the flight height of 30 m.

Finally get the shortest horizontal path: starting point $\rightarrow b_3 \rightarrow$ end point. The shortest horizontal path length is 2350.03 m.

4.  Flight zone (40 m, 50 m)

In the flight zone of (40 m, 50 m), the flight height is 40 m, the buildings no higher than 40 m will be abandoned, and these buildings are numbered 1, 2, 3, 4, 6, 7, 9, 10, 11, 12, 13, 14, 15, 16, 18, and 19.

The Voronoi diagram is used to generate 2 safe nodes. These points and their corresponding coordinates are shown in Table 6.

**Table 6.** Safe nodes at the flight height of 40 m.

| Safe Node | Coordinate (m) | Safe Node | Coordinate (m) |
| --- | --- | --- | --- |
| $a_4$ | (1304, 592) | $b_4$ | (1571, 39) |

The horizontal path obtained by the Dubins geometry path and Floyd algorithm is shown in Figure 13.

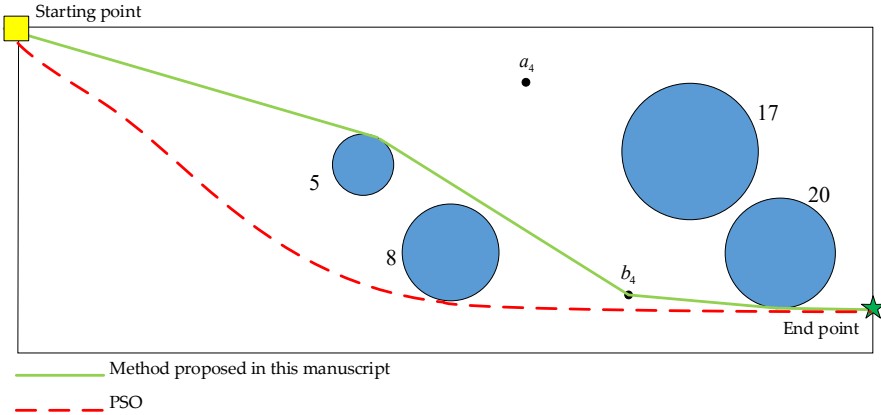

**Figure 13.** The horizontal path at the flight height of 40 m.

Finally, the shortest horizontal path and path length obtained are the same as those obtained when the flight height is 30 m, and the path length is also 2350.03 m. Since the safe node generated by buildings numbered 2 and 3 is located outside the rectangular area, they will not affect the path of the UAV after being removed

5.    Flight zone (50 m, 60 m)

In the flight zone of (50 m, 60 m), the flight height is 50 m. Only the building numbered 20 is reserved, and no safe node is generated. The horizontal path obtained by Dubins geometry path and Floyd algorithm is shown in Figure 14. The shortest horizontal path length is 2331.11 m.

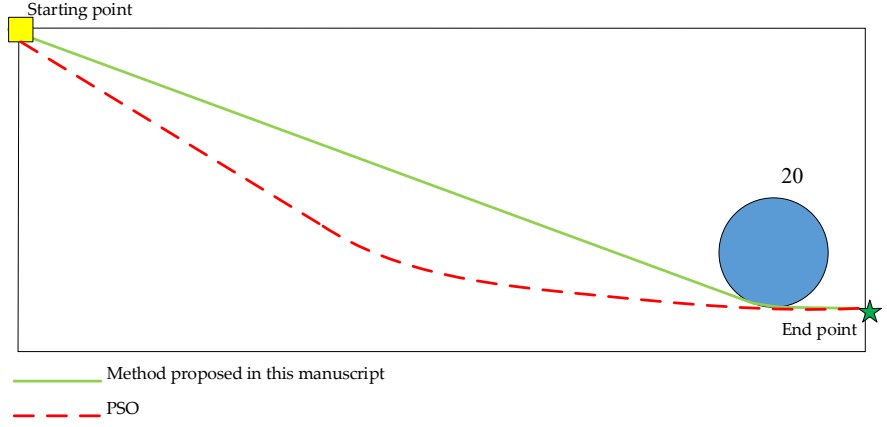

**Figure 14.** The horizontal path at the flight height of 50 m.

The iteration of PSO are shown in Figure 15.

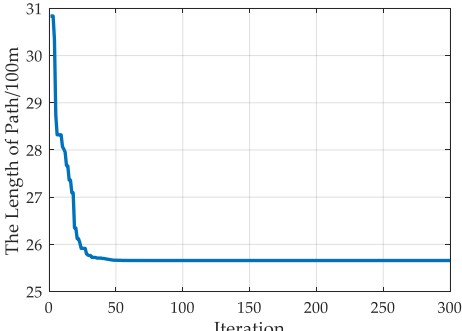

(**a**) Iteration at the flight height of 10 m

**Figure 15.** *Cont.*

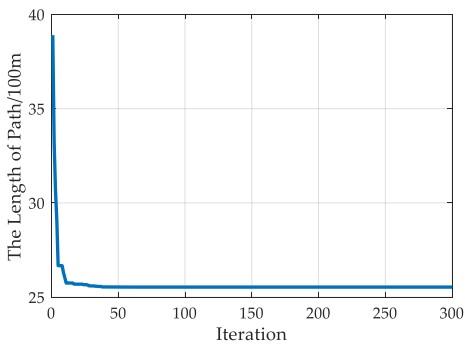

(**b**) Iteration at the flight height of 20 m

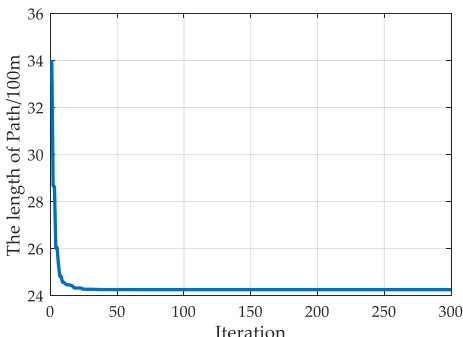

(**c**) Iteration at the flight height of 30 m

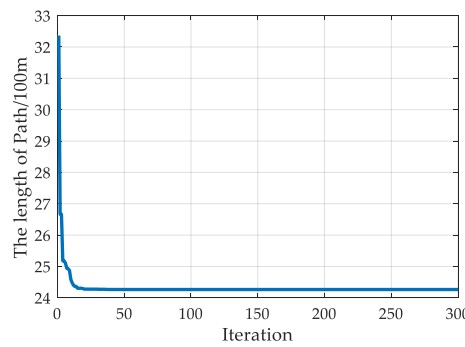

(**d**) Iteration at the flight height of 40 m

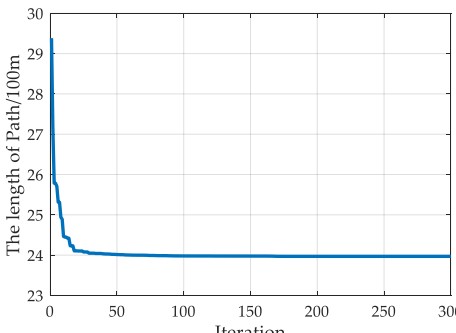

(**e**) Iteration at the flight height of 50 m

**Figure 15.** The iteration of PSO.

5.2.2. Comparison of Path Planning Methods

We compared the path length obtained by the path planning method in this study and PSO, the results are shown in the Table 7 and Figure 16.

**Table 7.** Comparison of path length.

| Flight Height (m) | Path Length Obtained by the Method in This Study (m) | Path Length Obtained by PSO (m) |
|---|---|---|
| 10 | 2392.82 | 2565.8 |
| 20 | 2357.32 | 2553.45 |
| 30 | 2350.03 | 2426.37 |
| 40 | 2350.03 | 2422.67 |
| 50 | 2331.11 | 2396.9 |

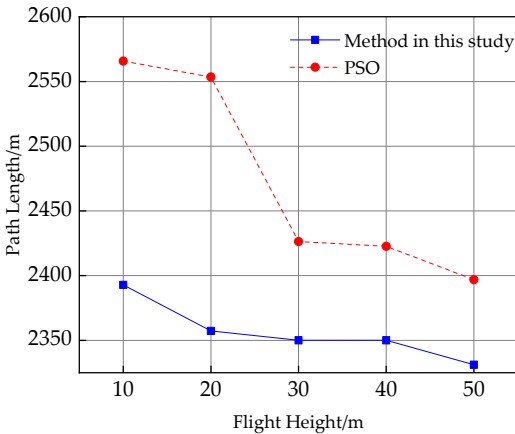

**Figure 16.** Comparison of path length.

From Table 7 and Figure 16, it can be concluded that the path length obtained by path planning method in this study is shorter than that obtained by PSO.

Compared with PSO, the path planning method proposed in this study has shorter length and more accurate calculation result, and the result obtained is the optimal path. Next, the horizontal path length obtained by the path planning method in this study will be used to calculate the battery energy consumption.

*5.3. Results and Analysis*

5.3.1. Low-Altitude Area and High-Altitude Area

According to Equations (6) and (7), it can be seen that the battery output power of eVTOL UAV is affected by the flight height, and also by the local altitude. According to the obtained flight height and horizontal flight path length, the energy consumption of eVTOL UAV can be calculated, and the energy consumption law of eVTOL UAV in low-altitude and high-altitude areas is studied and analyzed under standard atmospheric environment. The low-altitude area represented by Guangzhou city with an average altitude of about 10 m and the high-altitude area represented by Xining city with an average altitude of about 2260 m were selected.

In the low-altitude area with an altitude of 10 m, the calculation results are shown in Table 8.

**Table 8.** Calculation results in low-altitude areas.

| Flight Height (m) | Horizontal Path Length (m) | Battery Output Power in Horizontal Flight Phase (W) | Energy Consumption in Horizontal Flight Phase (J) | Energy Consumption in Take-off Phase (J) | Energy Consumption in Landing Phase (J) | Total Energy Consumption (J) |
|---|---|---|---|---|---|---|
| 10 | 2392.82 | 160.99 | 25,681.79 | 1069.23 | 1069.23 | 27,820.25 |
| 20 | 2357.32 | 160.99 | 25,300.34 | 2180.99 | 2180.99 | 29,662.32 |
| 30 | 2350.03 | 160.99 | 25,222.10 | 3272.27 | 3272.27 | 31,766.64 |
| 40 | 2350.03 | 160.99 | 25,221.93 | 4364.08 | 4364.08 | 33,950.09 |
| 50 | 2331.11 | 160.99 | 25,018.61 | 5456.41 | 5456.41 | 35,931.43 |

In the high-altitude area with an altitude of 2260 m, the calculation results are shown in Table 9.

**Table 9.** Calculation results in high-altitude areas.

| Flight Height (m) | Horizontal Path Length (m) | Battery Output Power in Horizontal Flight Phase (W) | Energy Consumption in Horizontal Flight Phase (J) | Energy Consumption in Take-Off Phase (J) | Energy Consumption in Landing Phase (J) | Total Energy Consumption (J) |
|---|---|---|---|---|---|---|
| 10 | 2392.82 | 164.62 | 26,260.40 | 1218.07 | 1218.07 | 28,696.54 |
| 20 | 2357.32 | 164.66 | 25,877.09 | 2436.76 | 2436.76 | 30,750.61 |
| 30 | 2350.03 | 164.69 | 25,801.76 | 3656.07 | 3656.07 | 33,113.90 |
| 40 | 2350.03 | 164.73 | 25,808.03 | 4875.99 | 4875.99 | 35,560.01 |
| 50 | 2331.11 | 164.76 | 25,604.91 | 6096.54 | 6096.54 | 37,797.99 |

According to the calculation results in the table, the variation rules of flight height and horizontal path length, horizontal flight output power, horizontal flight energy consumption, and the proportion of take-off and landing energy consumption in the total energy consumption in low and high-altitude areas are analyzed. The results are shown in Figure 17.

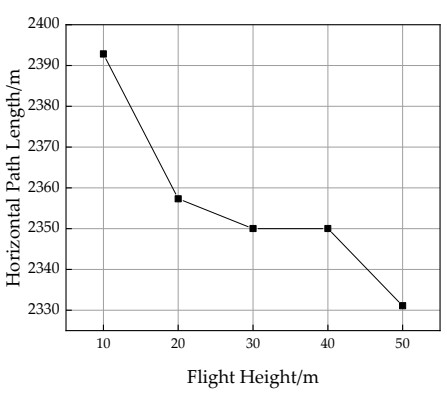

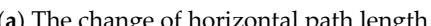

**(a)** The change of horizontal path length

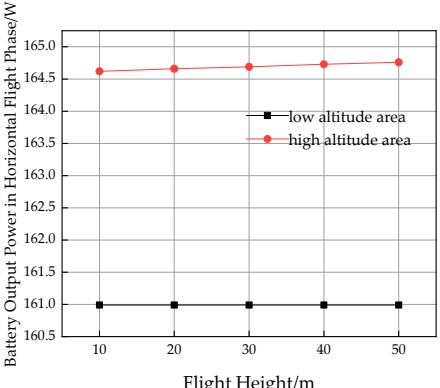

**(b)** The change of horizontal flight output power

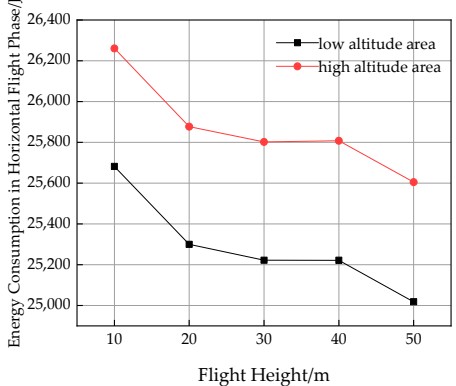

**(c)** The change of horizontal flight energy consumption

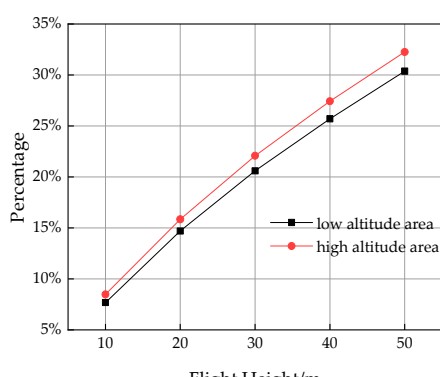

**(d)** The change of proportion of landing and takeoff energy consumption to total energy consumption

**Figure 17.** Changes in relationships.

In Figure 17a, the length of horizontal path generally decreases with the increase of flight height. The reason is that at higher altitudes, drones have fewer buildings to avoid and a shorter horizontal path length.

In Figure 17b, the output power of horizontal flight increases with the increase of flight altitude in both low-altitude and high-altitude areas, but the growth range is small. At

the same flight altitude, the output power of the horizontal flight in the high-altitude area is larger than that in the low-altitude area, indicating that the output power of the UAV battery will be larger in the high-altitude area, where the air is thinner and less dense.

In Figure 17c, the variation law of horizontal flight energy consumption in low-altitude areas and high-altitude areas is the same, and both of them generally show a downward trend with the increase of flight altitude. The reason is that the length of the horizontal path decreases with the increase of the flight altitude. Although the output power of the UAV battery increases, the growth is slow, which reduces the energy consumption of the horizontal flight.

In Figure 17d, the proportion of take-off and landing energy consumption in the total energy consumption increases with the increase of flight height, and the proportion reaches 30% at the highest level, and the proportion is larger in high-altitude areas. As the UAV flies in a small area of 2200 m × 730 m, the horizontal path length is short, resulting in a large proportion of energy consumption in the vertical flight phase in the total energy consumption.

### 5.3.2. Small Area and Large Area

In Section 5.3.1, the variation law of energy consumption of UAV in the small area is studied. The path planning of UAV in a small area is only applicable to the residential area. If the UAV flies in a larger area with a longer horizontal path, the proportion of take-off and landing energy consumption in the total energy consumption will be smaller, and the change rule of the total energy consumption may be different. According to the existing UAV battery performance, the UAV long distance transportation of 20 km to 30 km range, in order to analyze the energy consumption of the UAV flying in a larger area, now only to increase the horizontal path in Tables 8 and 9 to 10 times, flight height, flight speed, battery output power in horizontal flight phase, take-off and landing energy consumption are the same. The calculated results in large area are shown in Table 10. According to the total energy consumption data in Tables 8–10, Figure 18 is obtained.

**Table 10.** Calculation results in large area.

| Flight Height (m) | Horizontal Path Length (m) | Energy Consumption of Horizontal Flight at High-Altitude Area (J) | Energy Consumption of Horizontal Flight at Low Altitude Area (J) | Total Energy Consumption at High-Altitude Area/(J) | Total Energy Consumption at Low-Altitude Area/J (J) |
|---|---|---|---|---|---|
| 10 | 2392.82 | 262,604.00 | 256,817.90 | 265,040.20 | 258,956.30 |
| 20 | 2357.32 | 258,770.90 | 253,003.40 | 263,644.40 | 257,365.30 |
| 30 | 2350.03 | 258,017.60 | 252,221.10 | 265,329.30 | 258,765.60 |
| 40 | 2350.03 | 258,080.30 | 252,219.30 | 267,832.30 | 260,947.50 |
| 50 | 2331.11 | 256,049.10 | 250,618.10 | 268,242.20 | 261,098.90 |

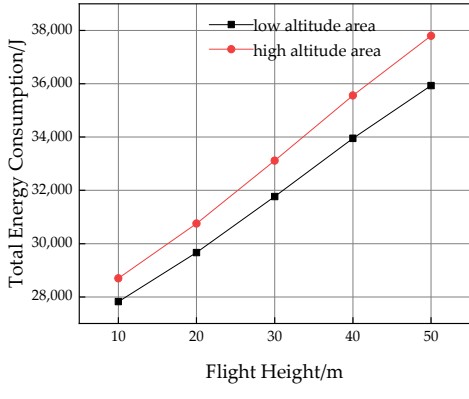

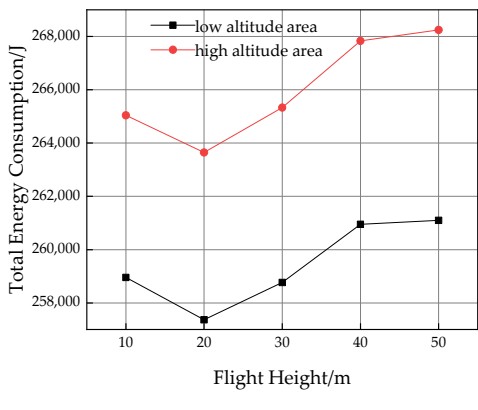

(**a**) Small area          (**b**) Large area

**Figure 18.** Variation law of total energy consumption.

*5.4. Discussion*

It can be seen from Table 10 and Figure 18 that the total energy consumption in low-altitude areas and high-altitude areas has the same change law. In the small area, since the take-off and landing energy consumption accounts for a large proportion of the total energy consumption, the total energy consumption in the low-altitude and high-altitude areas increases with the increase of the flight height. However, in a large regional environment, the proportion of take-off and landing energy consumption in total energy consumption decreases with the increase of horizontal path length, and the total energy consumption shows a fluctuating trend. The total energy consumption is minimum when the flight height is 20 m. According to the data in Table 10, there is a difference of 3733.9 J between the maximum and minimum total energy consumption in the middle and low-altitude areas in a large area. According to Table 8, the difference of energy can enable the UAV to cover an additional range of about 350 m. The difference between the maximum and minimum total energy consumption in the high-altitude area is 4598.2 J, corresponding to Table 9, it can be seen that the difference energy can make the UAV cover an additional range of about 420 m. In the future, a large number of UAVs will enter the urban airspace. If the flight height with the minimum energy consumption is selected, the transport range of UAVs can be increased. The energy saved by high-frequency flights at the same distance can save a considerable amount of money for UAV operators.

According to the above analysis, when UAVs are used to perform tasks in urban areas, UAV control strategies aiming at minimum energy consumption should be different in small and large areas. In small areas, fly the UAV as low as possible. However, in a large area, the total energy consumption does not necessarily increase with the increase of the flight altitude, so the flight altitude with the minimum energy consumption needs to be determined after calculation, analysis and comparison.

## 6. Conclusions

(1)   The eVTOL UAV combines the advantages of multi-rotor UAV and fixed-wing UAV. Its operation is characterized by vertical takeoff first and then switching to a fixed wing attitude during the horizontal flight phase. Therefore, we first consider the flight height of the UAV and then plan the horizontal path of the UAV.

The UAV horizontal path planning method proposed in this study can be applied to complex urban environment. In urban low-altitude airspace, the height, number and location of buildings affect the flight path of UAVs. The flight environment of UAVs is different at different flight heights, and the length of horizontal path decreases with the increase of height. The path planning method proposed in this study has shorter path distance and more accurate calculation process than the PSO, which can achieve the global optimal effect.

(2)   In both low-altitude and high-altitude areas, the proportion of take-off and landing energy consumption in the total energy consumption of UAV in the small area increases with the increase of flight height. The air at high altitudes is thinner and less dense. When flying at the same altitude, the battery output of a UAV at high altitudes is greater than that at low-altitudes.

(3)   In low-altitude and high-altitude areas, the total energy consumption of UAVs varies with flight altitude in the same way. However, in small and large areas, the variation of total energy consumption of UAV is different. In small area environment, the total energy consumption increases with the increase of flight altitude. In a large regional environment, the take-off and landing energy consumption takes a small proportion in the total energy consumption, and the total energy consumption fluctuates. Therefore, the energy consumption of the UAV at all flying altitudes should be analyzed and calculated, and the flying altitude with the minimum energy consumption can be obtained after comparison. The energy savings enable UAVs to cover longer distances, thus reducing operating costs for UAV operators.

In future work, we will consider the energy consumption of UAVs operating in wind field conditions. When the UAV flies through the wind field, the flight time increases and the battery energy consumption increases. Strong gusts can also cause changes in the attitude of the drone, which requires more power to ensure steady flight. The effect of this wind field on the drone would be far greater than the effect of changes in atmospheric density.

**Author Contributions:** All authors contributed equally to this study. Conceptualization, Y.L. and M.L.; methodology, M.L.; validation, M.L.; formal analysis, M.L.; resources, Y.L.; data curation, M.L.; writing—original draft preparation, M.L.; writing—review and editing, Y.L.; visualization, M.L.; supervision, Y.L.; funding acquisition, Y.L. All authors have read and agreed to the published version of the manuscript.

**Funding:** This research was funded by MOE (Ministry of Education in China) Youth Fund Project of Humanities and Social Sciences (Grant No. 21YJCZH075); General Program of Tianjin Applied Basic Research Diversified Investment Fund (Grant No. 21JCYBJC00720); Scientific Research Program of Tianjin Municipal Education Commission (Grant No. XJ2022003001).

**Institutional Review Board Statement:** Not applicable.

**Informed Consent Statement:** Not applicable.

**Data Availability Statement:** Not applicable.

**Conflicts of Interest:** The authors declare no conflict of interest.

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
