# Peer review of "Path Planning of Electric VTOL UAV Considering Minimum Energy Consumption in Urban Areas"

_sustainability, doi:10.3390/su142013421_

Round 1

Reviewer 1 Report

The manuscript "Path Planning of Electric VTOL UAV Considering Minimum Energy Consumption in Urban Areas" addresses an essential aspect of minimizing energy consumption for Electric VTOL UAV topic, in the broader UAS Air Traffic Management (UTM) area. This topic is in the journal's scope and is a current research focus.

The overall structure of the manuscript is optimal, and the style is clear, making the manuscript easily accessible and readable. The concept is sound, and the case studies were well conducted. Grammar and spelling are mostly correct, though some issues were found. Some careful proofreading is recommended.

Concerning the content, the authors are recommended to implement the following improvements:

1 This manuscript proposed a method of Path Planning for Electric VTOL UAV based on the Voronoi diagram, Dubins Geometric path, and Floyd algorithm. The authors also compared the proposed method with the PSO algorithm. The reviewer noticed that both methods, illustrated in this manuscript, adopt the strategy of first fixing the height and then planning the path. However, what about the strategy of considering horizontal and vertical planning at the same time? Could the author add some remarks or comments in the part of the Discussion or Conclusion?

2 The wind field conditions definitely affect the Path Planning proposed in this manuscript. The authors have mentioned such aspects in the Conclusions. However, could the authors describe it in a bit more detail, especially in the step of Dubins Geometric path. 

3 Minor typos/mistakes:

- Line 183, "2.1" should be "2.2"

- Line 563, "5.3.1" should be "5.3.2"

- Line 579, "5.1" should be "5.4"

- Line 604, "5 Conclusions" should be "6 Conclusions"

Reviewer 2 Report

1. The Abstract should be cut down. " The simulation results show that .... " is too long.

2. The Introduction should focus on the key problems about this paper, and should spend more words to show the research status about the path palnning or the energy consumption of the eVTOL UAV.

3. A pseudocode or flow diagram of the path planning method is necessary.

4. Which is the role of the PSO? The conmparison results should be presented in one figure.

Reviewer 3 Report

The manuscript studies the UAV path planning problem and the topic selection has certain theoretical significance and important practical reference value. And, the topic of the paper fall within the scope of this academic journal SUSTAINABILITY.

The reviewer suggests minor revision and I hope that my following comments will be helpful to the authors:

1. The manuscript analyzes the influence of flight height and altitude on the final result. Considering the strong correlation between flight height and altitude (with different reference planes), can the two be merged?

2. Does the model constructed by the author consider only one take-off and landing process in a mission, that is, vertical rise at the starting point and vertical landing at the endpoint?

3. The author has given the energy consumption of vertical take-off and landing and horizontal flight. I suggest the authors add a further balance analysis, that is, how many meters of vertical take-off and landing consumes the same amount of energy as how many meters of horizontal flight, which will help deepen the understanding of the energy consumption of UAVs.

4. For the calculation formulas (such as formula (3), etc.) of temperature, air density, and altitude changes in the manuscript, it is recommended to mark references.

5. Among the references in the manuscript, there are many references in Chinese, and it is recommended to cite some more references in English.

6. There are too many long sentences in the Abstract, it is recommended to use more short sentences.

Reviewer 4 Report

The work is interesting and provides new insights into the modern topic of path planning for UAVs in urban areas. Yet some corrections and details must be provided:

1. Please provide the objective function and subject to constraints for the optimization problem (... minimizing energy consumption...). And as a result, provide optimization statements.

2. In line 137 please correct the text (...Heuristic intelligent algorithms...), as the algorithms themselves can not be intelligent. Please add a description of what is intelligence in terms of optimization problems associated with the problem stated.

3. In line 212, please add the equation after the ":" sign, or rearrange the text, because now it seems that something is missing.

4. Figure 4 is called "Path optimization", yet the optimization terminology is not clear. Provide the path choosing mathematical criteria, subject function to be minimized, and the constraints of the function according to the methodology from the previous section.

5. Figure 6 title should be corrected because it is not a "Weighted network diagram", it is just a Network diagram. Either add weights for each path.

6. Please explain in more detail how can the proposed method choose paths if a large number of vehicles will be used on the same paths, compared to modern road jams.

7. Could this method be used in the agricultural and forestry sectors?

8. Please add grids to figures 20-22.

9. Does the structure and exterior of the UAV influence energy consumption?

Round 2

Reviewer 2 Report

1. The organization of this manuscript is illogical, and I can not get the key contribution. It has listed so many existing methods, but how to combine them? Where is the own work or contribution of the authors?

2. The abstract should be improved. I do not think the simulations can get so many conclusions. The abstract is different from the conclusions.

Reviewer 4 Report

The authors corrected all the needed parts of the manuscript.

Round 3

Reviewer 2 Report

The path planning results obatained by PSO and the proposed mathod should be presented in one figure to make a comparison. That means Figures 10 and 15 should be merged. The Section 5.2.2 should also be merged into Section 5.2.1.
